# Loyalty in Heritage Tourism: The Case of Córdoba and Its Four World Heritage Sites

**DOI:** 10.3390/ijerph17238950

**Published:** 2020-12-01

**Authors:** Juan Antonio Jimber del Río, Ricardo D. Hernández-Rojas, Arnaldo Vergara-Romero, Mª Genoveva Dancausa Dancausa Millán

**Affiliations:** 1Agricultural Economics, Finance and Accounting, Universidad de Córdoba, Avda. de Medina Azahara 5, 14071 Córdoba, Spain; jjimber@uco.es; 2Research Department, Ecotec University, Samborondón 092302, Ecuador; avergarar@ecotec.edu.ec; 3Research Department, Espiritu Santo University, Samborondón 092302, Ecuador; 4Management, Economic Applied and Stadistics, Universidad de Córdoba, Avda. de Medina Azahara 5, 14071 Córdoba, Spain; z62damim@uco.es

**Keywords:** loyalty, heritage tourism, management, cultural heritage

## Abstract

The aim of this research is to study visitor loyalty at a destination with heritage sites and to use the results to improve the competitiveness of the destination. This study used the SPSS AMOS software with a model of structural equations to evaluate the proposed hypotheses. A questionnaire was given to a sample of 428 tourists who visited the heritage sites in Córdoba. The management of any World Heritage City needs to know about the visitors’ experience at the destination, which includes their expectations for the trip, expected quality of the destination, satisfaction with the destination, and how these affect visitor loyalty to the city, because it is important to get the visitor to recommend, and return to, the destination. In the case of Córdoba, the research has proven that visitor loyalty depends on visitor satisfaction with the destination, which depends on the perceived quality and value of the visit. In addition, the following areas for improvement have been identified: improvement of the information about the destination, improvement of waiting times and the professionalization of specialized tour guides at heritage sites. Therefore, the findings are important for city managers in order to be able to take actions which increase the loyalty to, and competitiveness of, the city compared to other similar destinations with heritage sites.

## 1. Introduction

Heritage presents the historical culture of a destination and can be a way of attracting visitors to cities. International recognition by means of appointing World Heritage Sites by the United Nations Educational, Scientific and Cultural Organization (UNESCO) is a way of preserving and conserving these sites. An important consequence of these appointments is a significant increase in the number of visitors and tourists [1].

It can be very useful for cities that are World Heritage Sites to know the preconceived ideas that visitors have before visiting the city, along with how satisfied they feel with the trip when it is finished. This data can be used to continuously improve the tourist experience [2].

Knowing the visitors’ expectations for, satisfaction with, and rating of a region is important for the successful management of the region and can help managers prepare adequate plans for tourism [3].

Spain is rich in heritage as a result of its historical past. It has 48 sites included in the UNESCO World Heritage List (2019) and 13 in the Intangible Heritage List with 17,579 pieces of Cultural Interest of which 970 are historical sites (UNESCO, 2019). It is the third country, after Italy and China, with the largest number of cataloged sites. The plan to sustainably preserve, manage and promote all cities with historical heritage was initiated in 1993. The Group of Heritage Cities of Spain (GCPHE) is a group of fifteen Spanish cities. To be included in this group the city must have a historical site registered in the UNESCO World Heritage List. Córdoba stands out in this group for several reasons. First, it has one of the oldest inscribed heritage sites on the list in Spain, the Mosque−Cathedral, which is the second most visited heritage site in Spain. Córdoba also has four different sites included in the World Heritage List, which is unique in Spain, and finally it is a city where there was a historical coexistence of Arabic, Jewish and Christian cultures, which is reflected in the listed heritage sites.

The theoretical model used for this study is the American Customer Satisfaction Index (ACSI) [4], which has been widely used to study the tourism sector [5].

The ACSI model is used to identify the main influences on satisfaction and how better services can be provided [6]. ACSI takes customer satisfaction into account in three different ways, first with perceived quality [7,8,9], which is a measure of the consumer experience and has a direct, positive impact on the rating given for overall customer satisfaction. Second, perceived value [10] in the relationship between quality and the price paid. It is an unimportant factor for repeated purchases, but is important when the customer purchases for the first time. Finally, customer expectations [4], which are the customer’s experience before the purchase is made, and involve advertising or word of mouth comments. All three dimensions are expected to have a direct and positive influence on satisfaction, because if a customer values all three positively, it means they feel positive about their experience [11]. Loyalty is expected to be one of the main results of customer satisfaction [12].

Until now ACSI has been the most complete and effective theoretical model for customer satisfaction [13]. There are five structural variables in this model, customer satisfaction is the variable for the final result. Perceived quality, perceived value and customer expectations are the three background variables while customer loyalty is the result of the satisfaction variable. The model scientifically uses the process of consumer cognition and overall satisfaction in a system of cause-and-effect interactions with mutual influence and connection, which can explain the relationship between the consumption process and overall satisfaction.

The aim of this article is to contribute to the literature on the subject in two ways. Firstly, with a review of the variables that affect the loyalty of the heritage tourist and on a more practical level, to analyze the management of the site in order to improve the visitor experience at the cultural heritage site. To do this, an analysis of the expectations, satisfaction and loyalty of tourists visiting the World Heritage sites was carried out. The fieldwork was done in Córdoba. This study is one of the first to analyze the relationship between tourism and expectations and contributes to the decisions which should be taken for the successful management of tourism in the city with four sites in the UNESCO list of world heritage sites. In addition, this city was analyzed using a structural equations model. This study contributes to the literature and discusses the need for responsible tourism to contribute to the maintenance, development of heritage and the economy, especially the world heritage sites which are at risk from carbon footprints.

This article is structured in the following way; after this introduction, the second part explains the theoretical framework of the structural equations model, the third section presents the methodology used and the fourth section summarizes the results of the research. The conclusions of the study are then stated and discussed, followed by a list of the references used in the article.

## 2. Theorical Background

Cultural tourism is a rapidly growing area and heritage tourism has become a niche market [14]. Cities with consolidated cultural heritage are experiencing an exponential growth of this type of tourism and the number of visits to the city. Historical heritage and culture motivate tourists because of the perceptions they have of their own heritage and their desire to participate in a personal “historical heritage experience”. The diversity of culture offered in heritage cities attracts different groups of visitors with different profiles mainly because each destination has its own particular cultural offer [15]. This creates a competitive environment for attracting tourists between cities. In addition, tourists are motivated to visit a city for different reasons, such as the desire to learn more about the history of the place, participate in a recreational experience, the desire to be in contact with their own heritage and the perception that tourists have of a destination [16]. Therefore, tourism and visits to a heritage city should be understood as a social phenomenon [17] which is more than just visiting historical sites. Different specialized experiences are available for tourists to participate in, such as local customs, history, art, and the traditions and cultures of the city or area. Consequently, heritage tourism is important for cities that have heritage sites [18].

This study analyzes the relationships between the variables of perceived value, expected quality, perceived quality, satisfaction and loyalty, for visitors to heritage sites. Academic research in this area usually investigates the perceived quality of the service being studied [19]. In these studies, the researchers developed a methodology using perceived value, which was implemented for the first time, positioning perceived quality of the service as one of the most important factors for predicting the future behavior of the user. Perceived value is a reference to the general assessment by the tourist of the usefulness of the visit based on the perceptions received [20]. Cossío-Silva et al., 2018 [21] used this variable to realistically predict the possible future behavior of the tourist. These results are very useful for policy makers and service providers when deciding what and how to motivate tourists to visit a site or city. Perceived value has been analyzed for different types of tourism such as group trips [22], theme parks [23], festivals [24] or casinos [20].

There are numerous studies that confirm the relationship between satisfaction and quality [25,26,27] and confirm that they are key factors for the loyalty of the tourist [28].

A very important factor for any destination is that the visitor feels satisfied with the visit because the destination will then leave a positive memory of the visit, which increases their loyalty and therefore increases the competitiveness of the destination. Satisfaction can be defined as the overall evaluation of the services received by a tourist or visitor in comparison to the level of the services which the visitor expected to receive. Perceived quality is an important factor as it influences visitor satisfaction and therefore loyalty.

The relationship between satisfaction and loyalty has been studied by several authors in different areas [22,29], and recently for the telephone services sector [30]. Both variables are positively related showing that the probability of a visitor at a heritage site repeating or recommending the destination is high [25,31] when the visitor feels satisfied with the visit. Existing literature states that each tourist destination should prioritize achieving high levels of visitor satisfaction as a way to increase the number of visitors at a destination, which also has an effect on the competitiveness of the destination [32].

Local administration and the tourism industry must have information about the visitors to maintain or increase the appeal of a destination for visitors in order to increase both the number of days they stay and the amount of money they spend at the destination. Research using structural equations to study satisfaction and loyalty at heritage sites has not been widely studied, and this study is therefore a useful contribution to the existing literature.

### Turistification, Gentrification and Tourismphobia

In recent years, the cities that are declared world heritage sites are undergoing a process of touristification. This process often has a negative impact with the depopulation of popular parts of the city which are in touristic areas [33,34,35,36,37,38,39,40]. The depopulation of these areas brings with it the so-called gentrification process. After a process of urban transformation and the installation of new public security and cleaning services, these areas become fashionable and are demanded by tourists. Gentrification causes the displacement of the traditional population to other cheaper areas in the city [41,42,43,44,45]. This causes conflict in the price of real estate, the disappearance of traditional trade and small artisans which are replaced by large franchises and companies prepared for mass tourism, new ambiguously regulated economic activities (Airbnb). These all lead to the displacement of traditional residents in touristic areas. Recent studies into gentrification and touristification analyze the interactions between the growth in tourism and the urban transformation of the tourist destination. In Spain, academic articles have been published about the sharing economy and the tourism industry in Madrid and Barcelona [33,46,47], but there are almost no contributions about other territories or smaller touristic cities that can be deeply affected, such as the city of Córdoba with its four World Heritage sites. Tourism strategies, that revitalize historic centers are created as a way to rescue abandoned or depressed places. However, if these policies are poorly applied without taking into account the load capacity of these places, the city can become overloaded and world heritage cities can be turned into “theme parks” [48,49] which leads to problems of tourism management.

Scientific literature shows the negative effects of the above problem in big cities such as London, Paris, Prague, Madrid or Barcelona. The city of Barcelona in Spain attracts 76 million tourists per year, and there is now a new word in Spain, “tourism-phobia”. A law to curb tourism was passed in Venice, which has a total population of 55,000 people and an average of 60,000 visitors per day, with numbers increasing to 170,000 during Carnival. As a result, the residents of Venice are protesting about tourists with signs reading “Tourists go away” or “You are destroying this area” [50,51,52]. On the other hand, in a large number of cases (medium-sized cities and developing countries), the inhabitants of these communities depend on tourism, which gives them economic gains, employment and an income, with improved infrastructure, services, and standard of living [41,42,43,53,54] These communities may prefer to support the social costs of heritage tourism. [33,34,35,36,37,38,39,40,44,45].

Air pollution is another factor that can influence the decision to visit, the length of stay and even the rejection of destinations in World Heritage cities [55,56,57,58]. The city of Córdoba offers the visitor a wonderful opportunity to enjoy the city in this respect because it is a small city with little pollution, which is another necessary factor for a recommendable tourist destination of reference.

The most important World Heritage cities all have problems of massification [59,60,61,62,63,64,65,66,67], O’Reilly (1986) [68] was the first to define the term “tourist capacity” as the maximum number of tourists that a destination can contain. This may reflect a new component in the concept of tourism, which promotes rational growth to avoid excesses and a negative impact on the environment [69,70,71,72,73].

Knowing about the consequences of tourist massification, gentrification and tourism-phobia can help the public administration and policy makers to provide strategies and take action which can adapt to rapid changes in cities, avoid tourist bubbles [74], tourist massification and the depopulation of touristic parts of the city, avoiding the reduction in quality of heritage sites, enforcing responsible policies that balance the standard of living of residents with an adequate use of cultural heritage that contributes to the economic and social development of the city. This article is relevant in this context since it helps managers to take measures to properly manage tourism, taking into account the negative results that poor tourism management can have.

## 3. Research Objective, Methodology and Data

The variables proposed in this study to measure the loyalty of the visitors to the heritage city of Córdoba were, (1) the perceived quality of the city’s heritage, (2) the expected quality of the tourist’s visit to the world heritage city, (3) the perceived value, which is the positive or negative difference between the expected quality and the perceived quality, (4) satisfaction measured by the number of tourists who would not change their destination or who would be willing to pay more for the product or service received, and (5) the loyalty that tourists feel as a result of their satisfaction with the visit.

The following (Figure 1) hypotheses were formulated using the consulted literature:

**Hypothesis** **1** **(H1).**
*The quality expected by the visitor at the cultural heritage site significantly influences the visitor’s perceived quality of the site.*


**Hypothesis** **2** **(H2).**
*The quality expected by the visitor at the cultural heritage site significantly influences the visitor’s perceived value of the site.*


**Hypothesis** **3** **(H3).**
*The quality perceived by the visitor at the cultural heritage site significantly influences the visitor’s perceived value of the site.*


**Hypothesis** **4** **(H4).**
*The quality perceived by the visitor at the cultural heritage site significantly influences the satisfaction of the visitor with the site.*


**Hypothesis** **5** **(H5).**
*The value perceived by the visitor at the cultural heritage site significantly influences the satisfaction of the visitor with the site.*


**Hypothesis** **6** **(H6).**
*The satisfaction of the visitor at the cultural heritage site significantly influences the visitor loyalty to the site.*


### 3.1. Data

This study was carried out in Córdoba, Andalusia, which is in the south of Spain. The city of Córdoba is one of the most important in Spain due to its culture, tourism and heritage [75]. The data was collected with a questionnaire, which was given to tourists visiting the heritage sites in the city. To ensure the validity of the questionnaire, the questions used were based on those from similar previous studies [76]. Information was collected with a pilot sample and the internal consistency was rigorously analyzed. Items were also chosen from similar previous studies [77].

### 3.2. Methodology

The questionnaire consisted of five groups of questions, which were, (1) questions about expected quality, perceived quality and perceived value of the city in general, the safety and urban conservation, as well as transport, accommodation and catering, (2) questions about the visitor’s satisfaction with their trip to the city, (3) questions about the loyalty that visitors feel to the city, (4) questions about the city’s World Heritage monuments. (5) questions about the visitor’s demographic profile. Tourists were informed of the academic purposes and anonymity when responding. The consent to take the questionnaire was verbal. The anonymity of the respondent was guaranteed at all times. The questions in the first four parts of the questionnaire, about the expected quality, perceived quality, expected value, satisfaction with the destination and loyalty of the visitor, used a seven-point Likert scale, where one was the most negative answer and seven the most positive. Participation in the study was voluntary. The questionnaire was composed of 74 items grouped into the five areas above. The sample data was collected using a personal questionnaire at different times during the morning, afternoon and evening. The sample was chosen exclusively from tourists who had been staying for at least one week in the city whilst visiting it and its cultural heritage. The tourists were visiting the four world heritage sites in the City of Córdoba, which are, the Mosque−Cathedral, the Jewish Quarter of the Old Town, the Courtyards and the archaeological findings at Medina Azahara. These world heritage sites must be visited with enough time to fully appreciate and value them. At least a week should be taken to do this. For the fifth part, which was the sociodemographic profile and travel details, closed questions were used. The number of valid questionnaires was 428 in total, which gave a confidence level of 95% and a sample error of 3.25%.

Research data was tabulated and analyzed using IBM SPSS 23 statistical software (IBM Corporation, Armonk, NY, USA). Estimates of structural equations were completed with IBM SPSS Amos 23.

In Table 1, the unobserved and latent model constructs and the measurement errors of the proposed model are shown.

Generalized Least Squares (GLS) was used to estimate the final proposed model (Figure 2). The calculation included the terms of disturbance, regression coefficients, which represent the relationship between exogenous and endogenous latent variables, and the relationships between these variables.

Hu and Bentler (1995) [97] recommends using various indicators to evaluate the model fit. The chi-square ratio of the degrees of freedom (CMIN/DF) are suggested as indicators. The values of these goodness-of-fit statistics typically range from 0 to 1, with 1 indicating a perfect fit.

## 4. Results

The main findings of the fieldwork are described below. Firstly, the results of the descriptive analysis of the sociodemographic profile are shown, secondly, the reliability and validity of the proposed model, and finally, the analysis of the hypotheses.

The sociodemographic profile of the visitors to the city of Córdoba is shown in Table 2. It can be seen that 61% of the interviewees were women, compared to 39% men. The questionnaires were mainly answered by young people under the age of 30 (53%) with university-level education (62%).

In Table 3, we show the relationship between the observed and latent variables. The structural coefficients of the normalized model have also been calculated.

To endorse the goodness of fit of the model we propose, which supports the hypothesis we formulated, the measures for absolute, incremental and parsimonious fit were calculated.

### 4.1. Analysis of the Individual Reliability of the Items

The model ways analyzed in order to find the validity and reliability of the constructs and verify which are reflective and which are formative.

Formative constructs in the model (loyalty): these were evaluated using the recommendations of Sarstedt et al., 2019 [98]. Positive convergent validity tests were also made by investigating the redundancy of these constructs [99]. Five out of the six formative items in the proposed model (Table 4) had a value higher than 0.707 and the only variable with a value less than 0.707 exceeded 0.6 in a preliminary investigation. In all cases the weighting was nonzero. Therefore, all items can be maintained in the model, pending further analysis.

The individual reliability of the items can be analyzed using the reflective constructs. In order to do this the simple correlations of each observed variable with the respective constructs were analyzed. Carmines and Zeller (1979) [100] state that a minimum threshold of 0.707 is necessary for a variable to be accepted as part of a construct. The 62 reflective items have loads greater than 0.707, which indicates good reliability of the items that make up each first-order construct.

The validity and reliability of the constructs were then calculated before making conclusions about the relationships between them [101]. These tests were followed by an assessment of collinearity and checking that the variance inflation factor (VIF) < 5. The results showed no collinearity in the variables that were used for the loyalty construct.

The reflective constructions or B-mode of the model (satisfaction, expectations and quality) were also analyzed (Table 5). The individual reliability of the indicators, the internal consistency of the construct, the convergent validity and the discriminant validity of the reflective constructs were all calculated [102]. The results indicated adequate individual reliability, as all load values were above the required minimum threshold of 0.505 [103] or 0.6 [101]. In fact, the analysis revealed that the loads were statistically significant to 99.99%. Based on the results of these calculations, the measurement model was considered valid and reliable, which meant that the structural model could then be analyzed.

### 4.2. Analysis of the Reliability of the First-Order Constructs

The values of Cronbach’s Alpha Coefficient and Composite Reliability indicate whether the observed variables rigorously measure the latent variable which they represent. According to Nunnally and Bernstein (1978) [104], the minimum value for acceptable Composite Reliability is 0.7. The values of composite reliability in Table 6 can be seen to all be above 0.7. The reliability of the first-order constructs or dimensions, and their capacity for measuring Loyalty has therefore been shown. In this study, 3 of the 5 constructs exceeded a value of 0.9 (satisfaction, expected quality and perceived quality) and 2 constructs exceeded 0.85 (loyalty and perceived value), which means that there can be no doubt about the constructs capacity for measuring loyalty.

### 4.3. Convergent Validity

Convergent validity analyzes whether a set of variables really evaluates a particular construct and not a different one. Average variance extracted (AVE) is the criterion for acceptance which is most commonly used in research (Table 7). Fornell and Larcker (1981) [103] stated that the value of AVE must be greater than 0.5, meaning that the construct shares more than half of its variance with its indicators, with the rest being the variance due to the measurement error [98]. This criterion applies only to latent variables with reflective indicators or for second-order constructs. The nine dimensions in the loyalty scale share more than 50% of their variance with their items. Therefore, this parameter is at an acceptable level.

### 4.4. Hypothesis Testing

In order to verify the goodness-of-fit of the proposed model and show that the hypotheses are accepted, the values of the measurements for absolute, incremental and parsimony fit were calculated. These are shown in Table 8 below.

The results obtained for the measurements show that the model has an acceptable goodness-of-fit for all the types of measurements, which, together with the importance of the model coefficients, justifies their validity and applicability.

The significance of the path coefficient of each hypothesis was then calculated (Table 9). This showed that five hypotheses are supported (H1, H2, H3, H4, and H6) and one hypothesis is not supported (H5).

Figure 3 shows the latent variables with the values of the structural coefficients. The limit probability of each variable observed for each latent variable is also shown. This probability is used to validate the importance of the relationships between the constructs of the proposed model.

This means that Hypotheses H1 and H2 (expected quality–perceived quality/perceived value), H3 (there is a positive relationship between perceived quality–perceived value), H4 (there is positive relationship between perceived quality−satisfaction) and H6 (there is positive relationship between satisfaction−loyalty) were all supported. Hypothesis H5 was not supported, which means that the satisfaction of visitors to heritage sites does not influence their feelings of loyalty to the site.

## 5. Discussion

Visitor loyalty is a key element for improving the competitiveness of tourist destinations, and therefore the variables that affect loyalty also influence the improvement. This study confirmed most of the hypotheses formulated and these can be used to improve the competitiveness of a destination. This study created a structural equation model that was used to study the loyalty of the heritage tourist and the relationships with the quality, perceived value and satisfaction with the site.

The variables that influenced visitors to choose a world heritage city as a destination were the quality that the tourist expected to find and the loyalty of third parties to the destination [21,105,106]. Once the visitor experience was over, the overall perceived value, which is a comparison of the expected and perceived value, gave a feeling of satisfaction which made visitors recommend it as a destination to others.

Hypothesis 1 was confirmed and showed that the expected quality of a destination directly influences the perceived quality. This relationship between expected and perceived quality has previously been observed by other authors [107,108]. To take advantage of this finding, managers in the tourism sector must promote the high-quality image of the destination in targeted tourism sectors. Hypothesis 2 was also confirmed, which showed that the quality expected by a visitor before visiting a destination directly influenced the final perceived value of the visit. This relationship has been previously studied and confirmed in studies in other cities [109]. Loyalty at destinations with a strong heritage component is determined by the expected and perceived quality of the heritage site [110].

Hypothesis 3 was also confirmed and showed the direct influence that the perceived quality of a destination has on the perceived value of the destination [111,112]. This finding coincided with that of other studies [83]. This means that there must be sufficient, interesting and easily accessible information about the heritage destination available to the visitor. Hypothesis 4 was also confirmed and demonstrates the direct influence that the perceived quality of a destination has on the satisfaction of the visitor. A high level of perceived quality can be expected to have a significantly positive effect on the visitor’s satisfaction at a destination [105,113,114]. In practice, this means that cities must take advantage of the tourist resources that increase the perceived value of the destination by tourists. Córdoba has a large number of different cultural treasures from the different cultures that have passed through this millennial city throughout history. The Synagogue of Jewish Culture, the Mosque and the archaeological site at Medina Azahara of Arab culture and the Alcazar of the Christian kings of Catholic Christian culture should be used to make tourists experience and feel that they are in a unique place and that the money paid for the visit to this city has been well-spent.

Hypothesis 5 was not confirmed, which means that the visitor’s perceived value of the heritage site does not significantly influence their satisfaction. This result had also been found in the research by [115]. The results of this study show that the perceived quality and the expected quality have a direct influence on satisfaction. The relationship of perceived value and satisfaction is not supported. One of the reasons for this is because satisfaction is influenced by the expectations created before the trip, the expectations that the visitor has after reading about the destination, planning itineraries, hotels, gastronomy, etc., and once the visitor is at the destination and has tried the gastronomy, s/he decides on the level of satisfaction with the visit and recommends it if the level of satisfaction is high [116,117]. The main reason why hypothesis 5 is not supported is due to the idiosyncrasies of this type of tourism. The results obtained coincided with studies that have analyzed this type of tourism and the relationships between some of the factors analyzed in this study. It was found that the motivation of the tourist can influence their satisfaction and loyalty. When the motivation for the visit is due to cultural heritage, both the expectations and the expected quality for the visit take second place [118,119] as the most important thing for the tourist motivated by cultural heritage is the visits to the sites, which alters the results for our hypotheses. Cultural and heritage motivation is so influential in this type of tourism that the research hypothesis of this study is not fulfilled as perceived value does not directly influence satisfaction. However, according to other authors, perceived value does influence other variables such as loyalty [120,121]. Therefore, even though the hypothesis is not fulfilled, the variable should still be taken into account by public managers. This means that cities that attract visitors because of important heritage sites should research satisfaction rather than perceived value to establish a long-term relationship with visitors and tourists.

Finally, Hypothesis 6 was confirmed, which shows the direct and positive influence of the satisfaction of the heritage tourist affects the loyalty of the heritage tourist. A high level of satisfaction is expected to have a positive effect on the loyalty of the heritage tourist to a heritage site. Based on previous research [114,122,123] the way in which the factors of satisfaction, quality and expectations influence the visitors’ intention to return to or recommend the destination were investigated. This study confirmed the relationship of high visitor satisfaction with the heritage of Córdoba and the willingness of tourists to return and recommend the city as a destination. This means that the loyalty of heritage tourists to a heritage destination is based on factors such as, safety at the destination and its surroundings, the accommodation, gastronomy, transport and the management of information provided to visitors, along with the comfort felt and the upkeep of the destination [124]. The visitor should feel that all this makes the tourist experience worth the price paid. In addition to enjoying visits to heritage sites, tourists also appreciate special rates, offers and discounts during their visit [125].

## 6. Conclusions

Sustainable heritage tourism aims to reach a balance between the traditional theory of social exchange, meaning compensating economic costs with profits, that uses the latent variable of perceived value, and the new environmental paradigm, which highlights the conservation/preservation of all resources and works for an improvement of the well-being of communities for the following generations [13,126]. Therefore, it must be taken into account for the management of heritage sites. The perceived value of a site by tourists influences tourist satisfaction and loyalty. World Heritage cities and especially the city of Córdoba with four cultural world heritage sites, have to be a reference for sustainable tourism that contributes to the maintenance of this heritage and Sustainable Development Goals.

This aim of this study is to provide the managers of the heritage sites in Córdoba an indication of what is needed to provide a satisfying visitor experience for those who visit this city. This means that the tourist satisfaction with the visit must be one of the reasons to exhibit the heritage at the destination. In agreement with previous studies [31], the results of this study indicate that visitor satisfaction has a positive influence on loyalty to the destination, which encourages tourists to return to the destination in the future, and to recommend it to others once they return to home. This study finds the most influential aspects for increasing the loyalty of heritage tourists in a city through which different cultures have passed and left important heritage sites that reflect the culture that inhabited the city.

As the results show, the tourist perceives unique values during their visit as a result of experiencing the different cultures present in the city. The preservation and cleanliness of the heritage sites were pointed out, along with accessibility in a friendly historic center were all considered advantages for visitors [127,128]. In this way, Córdoba unites and links the tourist with the destination, positively influencing loyalty to the city. The brand achieved by this set of heritage sites combined with the perceived quality of the visit is a recognized attraction for visitors who want to get to know the local culture during their visit [89].

The limitations of this study are due to the sample. The data was obtained from heritage tourists in Córdoba, which means that the collected data is valid for a certain type of tourist at a certain type of destination. It would be interesting to study a larger sample that includes other heritage cities and other countries in order to generalize and contrast results. It can also be seen that this study gives data and results for one particular moment in time. For a further reaching dataset and results, a longitudinal study must be implemented in order to collect the general perception of tourists over time. In this way evolution of the tourist perceptions and feelings could be analyzed over time. Similarly, measuring loyalty with intention of future behavior is a limitation of the study. Although it has been used in previous studies, it is still a subjective measure of behavior and does not always correspond to real behavior [122].

## Figures and Tables

**Figure 1 ijerph-17-08950-f001:**
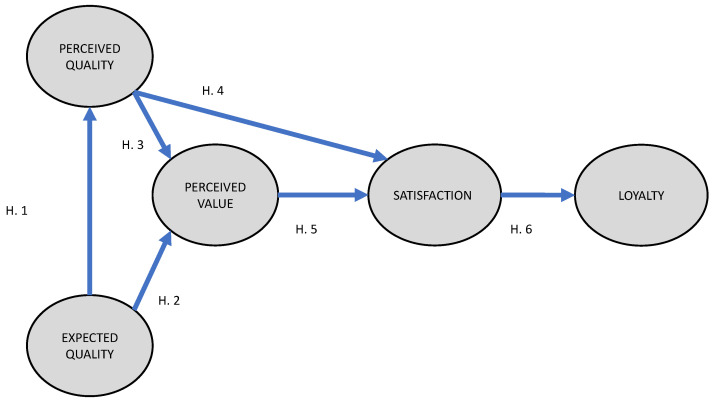
Research model.

**Figure 2 ijerph-17-08950-f002:**
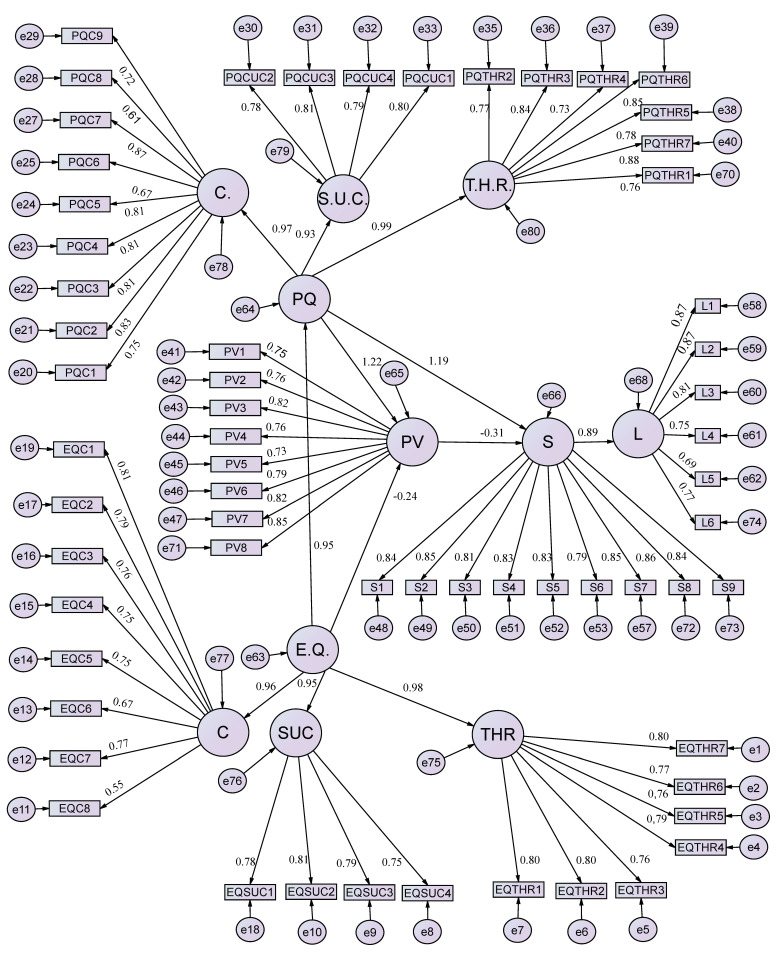
Path diagram of the proposed model.

**Figure 3 ijerph-17-08950-f003:**
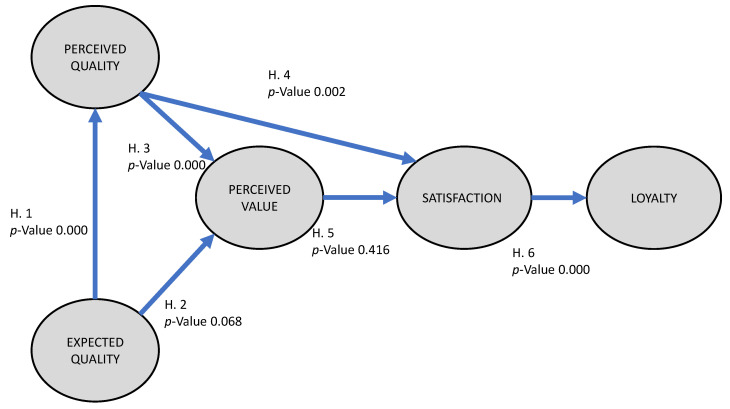
Path diagram of the proposed model with the *p* value.

**Table 1 ijerph-17-08950-t001:** Scales used.

Reference	Dimension	Indicators
		(EQC1) City prestige expectations, (EQC2) City cultural wealth expectations, (EQC3) City leisure offer expectations, (EQC4) Hospitality/treatment expectations, (EQC5)City cultural activities and entertainment expectations, (EQC6) City overcrowding expectations, (EQC7) Expectations for commercial supply, (EQC8) Expectations for waiting time for monuments, (EQSUC1) Expectations for information and signage, (EQSUC2) Expectations for safety in general, (EQSUC3) Expectations for urban cleaning, (EQSUC4) Expectations for urban environment conservation, (EQTHR1) Expectations of catering on offer, (EQTHR2) Expectations of gastronomic richness, (EQTHR3) Expectations of quality of the restaurant service, (EQTHR4) Expectations of accommodation on offer, (EQTHR5) Expectations of quality of service of accommodation, (EQTHR6) Expectations of communications to reach the city, (EQTHR7) Expectations of communications in the city

[78,79,80,81]	Expectative quality(EQ)


[25,31,82,83,84,85]	Satisfaction(S)	(S1) Satisfaction of stay in the city, (S2) Satisfaction meets expectations, (S3) Satisfaction of professional hospitality and catering treatment, (S4) Satisfaction of cultural heritage, (S5) Satisfaction of gastronomy offer, (S6) Satisfaction of professional hotel treatment, (S7) Satisfaction with cultural experience,(S8) Satisfaction with gastronomic experience,(S9) General satisfaction with city visit
[86,87,88,89]	Loyalty(L)	(L1) Loyalty would recommend friends, (L2) Loyalty would recommend family, (L3) Loyalty information about the city, (L4) Loyalty would return on the next vacation, (L5) Loyalty would return in the short term, (L6) Loyalty would return in the long term.
[90,91,92]		(PQC1) Perception of city prestige, (PQC2) Perception of city cultural richness, (PQC3) Perception of city leisure offer, (PQC4) Perception of hospitality/treatment, (PQC5) Perception of cultural activities and city shows, (PQC6) Perception of city overcrowding, (PQC7) Perception of commercial outlets on offer, (PQC8) Perception of waiting time for monuments, (PQSUC1) Perception of information and signage, (PQSUC2) Perception of security in general, (PQSUC3) Perception of urban cleanliness, (PQSUC4) Perception of conservation of urban environment, (PQTHR1) Perception of the catering on offer, (PQTHR2) Perception of gastronomic wealth, (PQTHR3) Perception of the quality of the restaurant service, (PQTHR4) Perception of the accommodation on offer, (PQTHR5) Perception of the quality of the service of accommodation, (PQTHR6) Perception of communications reaching the city, (PQTHR7) Perception of communications in the city.

Perceived quality(PQ)


[93,94,95,96]	Perceived value(PV)	(PV1) Perceived value of the leisure on offer, (PV2) Perceived value of cultural activities, (PV3) Perceived value of the commercial outlets on offer, (PV4) Perceived value of conservation of monuments, (PV5) Perceived value of the restoration, (PV6) Perceived value of the accommodation on offer, (PV7) Perceived value of communications to get to the city, (PV8) Perceived value of communications in the city

**Table 2 ijerph-17-08950-t002:** Sociodemographic profile of visitors.

Variable	Category	Absolute Frequency	Percentage
Sex (*n* = 428)	Male	166	38.8
	Female	262	61.2
Age (*n* = 428)	[less than 30]	228	53.3
	[30,31,32,33,34,35,36,37,38,39]	62	14.5
	[40,41,42,43,44,45,46,47,48,49]	68	15.9
	[50,51,52,53,54,55,56,57,58,59]	56	13.1
	60 or more	14	3.3
Studies (*n* = 428)	No studies	5	1.2
	Junior school	35	8.2
	Secondary school	124	38.3
	University	264	61.7

**Table 3 ijerph-17-08950-t003:** Standardized structural coefficients of the observed variables.

Latent Variable	Observed Variable	Standardized Coefficient	*p*-Value
EXPECTATIVE QUALITY	(EQC1) Prestige expectations of the city	0.806	***
	(EQC2) Expectations of cultural wealth of the city	0.786	***
	(EQC3) City leisure offer expectations	0.760	***
	(EQC4) Expectations hospitality/treatment	0.746	***
	(EQC5) Expectations for cultural activities and city shows	0.750	***
	(EQC6) Expectations of degree of overcrowding in city	0.675	***
	(EQC7) Expectations of commercial outlets on offer	0.767	***
	(EQC8) Expectations of waiting time for monuments	0.552	***
	(EQSUC1) Information and signaling expectations	0.782	***
	(EQSUC2) General safety expectations	0.815	***
	(EQSUC3) Expectations of urban cleaning	0.794	***
	(EQSUC4) Expectations of conservation urban environment	0.754	***
	(EQTHR1) Expectations for catering on offer	0.804	***
	(EQTHR2) Expectations for gastronomic richness	0.797	***
	(EQTHR3) Expectations of quality of service restoration	0.758	***
	(EQTHR4) Expectations of accommodation on offer	0.787	***
	(EQTHR5) Expectations of quality service accommodation on offer	0.761	***
	(EQTHR6) Expectations of communications to reach the city	0.770	***
	(EQTHR7) Communications expectations in the city	0.797	***
LOYALTY	(L1) Loyalty I would recommend to friends	0.868	***
	(L2) Loyalty I would recommend to family	0.872	***
	(L3) Loyalty city information	0.814	***
	(L4) Loyalty would return on the next vacation	0.745	***
	(L5) Loyalty would return in the short term	0.689	***
	(L6) Loyalty would return in the long run.	0.766	***
PERCEVED QUALITY	(PQC1) Prestige perception of the city	0.832	***
	(PQC2) Perception of cultural wealth of the city	0.748	***
	(PQC3) Perception of city leisure offer	0.810	***
	(PQC4) Hospitality/treatment perception	0.811	***
	(PQC5) Perception of cultural activities and city shows	0.808	***
	(PQC6) City overcrowding degree perception	0.681	***
	(PQC7) Perception of commercial outlets on offer	0.823	***
	(PQC8) Perception of waiting time for monuments	0.611	***
	(PQC9) Perception of conservation of monuments	0.780	***
	(PQSUC1) Information perception and signaling	0.810	***
	(PQSUC2) Perception of safety in general	0.793	***
	(PQSUC3) Perception of urban cleaning	0.765	***
	(PQSUC4) Perception of urban environment conservation	0.843	***
	(PQTHR1) Perception of catering on offer	0.725	***
	(PQTHR2) Perception of gastronomic wealth	0.776	***
	(PQTHR3) Perception of quality of restaurant service	0.848	***
	(PQTHR4) Perception of accommodation on offer	0.879	***
	(PQTHR5) Perception of quality of service of accommodation on offer	0.759	***
	(PQTHR6) Perception of communications to reach the city	0.716	***
	(PQTHR7) Perception of communications in the city	0.804	***
PERCEVED VALUE	(PV1) Perceived value leisure on offer	0.758	***
	(PV2) Perceived value cultural activities	0.751	***
	(PV3) Perceived value commercial outlets on offer	0.823	***
	(PV4) Perceived value conservation of monuments	0.756	***
	(PV5) Perceived value restaurants on offer	0.734	***
	(PV6) Perceived value accommodation on offer	0.790	***
	(PV7) Perceived value of communications to get to the city	0.824	***
	(PV8) Perceived value of communications in the city	0.854	***
SATISFACTION	(S1) Satisfaction of stay in the city	0.835	***
	(S2) Satisfaction meets expectations	0.852	***
	(S3) Satisfaction of professional hospitality and restaurant treatment	0.811	***
	(S4) Satisfaction of cultural heritage	0.826	***
	(S5) Satisfaction of gastronomy on offer	0.827	***
	(S6) Satisfaction of professional hotel treatment	0.794	***
	(S7) Satisfaction of cultural experience	0.855	***
	(S8) Satisfaction of gastronomic experience	0.858	***
	(S9) General satisfaction of visit to the city	0.835	***

*** *p*-Value > 0.001

**Table 4 ijerph-17-08950-t004:** Reliability of individual indicators (Formative).

Observed Variable	Weighting
L1	0.813
L2	0.846
L3	0.765
L4	0.819
L5	0.860
L6	0.687

**Table 5 ijerph-17-08950-t005:** Individual reliability of the indicators (reflective).

Variable	Weighting
S1	0.689
S2	0.678
S3	0.624
S4	0.682
S5	0.684
S6	0.582
S7	0.739
S8	0.700
S9	0.687
EQC1	0.605
EQC2	0.575
EQC3	0.671
EQC4	0.462
EQC5	0.610
EQC6	0.679
EQC7	0.562
EQC8	0.502
EQSUC1	0.583
EQSUC2	0.581
EQSUC3	0.540
EQSUC4	0.657
EQTHR1	0.622
EQTHR2	0.613
EQTHR3	0.567
EQTHR4	0.577
EQTHR5	0.503
EQTHR6	0.507
EQTHR7	0.511
EQTHR8	0.548
PV1	0.568
PV2	0.535
PV3	0.558
PV4	0.480
PV5	0.456
PV6	0.488
PV7	0.568
PV8	0.523
PQC1	0.585
PQC2	0.683
PQC3	0.626
PQC4	0.587
PQC5	0.632
PQC6	0.530
PQC7	0.580
PQC8	0.327
PQC9	0.433
PQCUC1	0.515
PQCUC2	0.464
PQCUC3	0.583
PQCUC4	0.615
PQTHR1	0.499
PQTHR2	0.685
PQTHR3	0.642
PQTHR4	0.482
PQTHR5	0.507
PQTHR6	0.513
PQTHR7	0.491

**Table 6 ijerph-17-08950-t006:** Composite Reliability and Cronbach’s Alpha.

Construct		Composite Reliability	Cronbach’s Alpha
L	Loyalty	n/a	0.859
S	Satisfaction	0.705	0.939
EQ	Expected quality	0.703	0.936
PV	Perceived value	0.716	0.869
PQ	Perceived quality	0.701	0.930

**Table 7 ijerph-17-08950-t007:** Average variance extracted.

Construct		Average Variance Extracted (AVE)
L	Loyalty	n/a
S	Satisfaction	0.586
EQ	Expectative quality	0.643
PV	Perceved value	0.620
PQ	Perceved quality	0.693

**Table 8 ijerph-17-08950-t008:** Goodness-of-fit.

Goodness-of-Fit Measurement	Value
Absolute Fit	
Chi-squared/DF	1.852
GFI	0.745
RMSEA	0.045
RMR	0.132
Incremental Fit	
AGFI	0.727
Parsimony Fit	
PNFI	0.119
PCFI	0.200

**Table 9 ijerph-17-08950-t009:** Hypothesis testing.

Hypothesis	Effect	Path Coefficient	*p*-Value	Supported?
H1: Expected quality–perceived quality	+	0.868	0.000 ***	YES
H2: Expected quality–perceived value	−	0.253	0.068 *	YES
H3: Perceived quality–perceived value	+	1.387	0.000 ***	YES
H4: Perceived quality–satisfaction	+	1.377	0.002 **	YES
H5: Perceived value–satisfaction	−	−0.315	0.416	NO
H6: Satisfaction–loyalty	+	0.896	0.000 ***	YES

a = 0.001 (***), a = 0.01 (**), a = 0.05 (*).

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
