# Peer review of "Loyalty in Heritage Tourism: The Case of Córdoba and Its Four World Heritage Sites"

_ijerph, 2020, doi:10.3390/ijerph17238950_

Round 1

Reviewer 1 Report

The article presents an important advance of knowledge about the subject matter.

The topic is attractive and the research is relevant.

There is coherence between the research design and the problem studied. Likewise, the analysis and discussion of the data is solid.

The article focuses very well on the subject and justifies the object of study. The importance and interest of the problem is observed.

Requires minor revisions:

You need also study more literature on loyalty, heritage tourism and management. Most of those weaknesses are connected with insufficient literature review.  In this area, the problem lies in the fact, that what you present as your outcome is not much more than what is known for many years and treated as obvious.

Author Response

MANUSCRIP ID: ijerph-1001603: Loyalty in heritage tourism: the case of Córdoba and its four world heritage sites.

Dear editor and reviewers, thank you very much for allowing us revising and resubmitting tour article to the International Journal of Environmental Research and Public Health. We have found your comments to be highly helpful in improving the article in terms of the introduction, the theoretical background, the research methodology, and the theoretical contributions. After carefully reading your comments, we have introduced some changes in the manuscript to address your concerns. Following your recommendation, as shown in the new version of the manuscript. We present our detailed comments below.

RESPONSE TO REVIEWER 1

1) Requires minor revisions:

You need also study more literature on loyalty, heritage tourism and management. Most of those weaknesses are connected with insufficient literature review.  In this area, the problem lies in the fact, that what you present as your outcome is not much more than what is known for many years and treated as obvious.

Thank you very much for your constructive suggestion; it has led us to re-write the "Theoretical foundation: loyalty, heritage tourism and managament".  The authors have added the serial number six and seven:

“…first with Perceived Quality [7-9] ], which is a measure of the consumer experience and has a direct, positive impact on the rating given for overall customer satisfaction.  Second, Perceived Value [10] ] in the relationship between quality and the price paid. It is an unimportant factor for repeated purchases, but is important when the customer purchases for the first time. Finally, customer expectations [4], which are the customer’s experience before the purchase is made, and involve advertising or word of mouth comments.  All three dimensions are expected to have a direct and positive influence on satisfaction, because if a customer values all three positively, it means they feel positive about their experience [11]. Loyalty is expected to be one of the main results of customer satisfaction [12].”

Reviewer 2 Report

The aim of the research is to increase the conclusions already obtained from the reference literature from which the research itself starts. More specifically, this study aims to build a model of reliable surveys about the expectations and perceptions of tourists who visit a Unesco heritage site.

The accurate verification of the methods used is a central concern of the authors who hope that this model can be a useful tool for the public administrators of sites home to assets included in the UNESCO heritage but not necessarily, as it can be useful in general for the administrators of localities with heritage of cultural tradition.
This verification takes up a large part of the essay and seems very accurate, even in its graphic and expository presentation, even if the undersigned is not able to really evaluate its technical consistency, being totally inexperienced on the subject.
Thanks to the data collected and so diligently analyzed, the authors refute or accept some hypotheses put forward by the international literature which appears to be rich and comprehensive. Although - as the authors themselves acknowledge - other studies are needed that implement the data collected especially on a longer time scale, the fundamental aim of this research can be said to be achieved.

Author Response

MANUSCRIP ID: ijerph-1001603: Loyalty in heritage tourism: the case of Córdoba and its four world heritage sites.

Dear editor and reviewers, thank you very much for allowing us revising and resubmitting tour article to the International Journal of Environmental Research and Public Health. We have found your comments to be highly helpful in improving the article in terms of the introduction, the theoretical background, the research methodology, and the theoretical contributions. After carefully reading your comments, we have introduced some changes in the manuscript to address your concerns. Following your recommendation, as shown in the new version of the manuscript. We present our detailed comments below.

RESPONSE TO REVIEWER 2

1) The aim of the research is to increase the conclusions already obtained from the reference literature from which the research itself starts. More specifically, this study aims to build a model of reliable surveys about the expectations and perceptions of tourists who visit a Unesco heritage site.

The accurate verification of the methods used is a central concern of the authors who hope that this model can be a useful tool for the public administrators of sites home to assets included in the UNESCO heritage but not necessarily, as it can be useful in general for the administrators of localities with heritage of cultural tradition.

This verification takes up a large part of the essay and seems very accurate, even in its graphic and expository presentation, even if the undersigned is not able to really evaluate its technical consistency, being totally inexperienced on the subject.

Thanks to the data collected and so diligently analyzed, the authors refute or accept some hypotheses put forward by the international literature which appears to be rich and comprehensive. Although - as the authors themselves acknowledge - other studies are needed that implement the data collected especially on a longer time scale, the fundamental aim of this research can be said to be achieved.

Thank you very much for the revision made. The authors agree on their comments in particular we greatly appreciate that it is a useful tool for the management of tourism in regions that have goods on the world heritage list.

Reviewer 3 Report

The article refers to a robust area of research on destination image destination, and especially the relationship between the quality assessment and satisfaction of visitors and destination loyalty in the context of world heritage sites. The authors used the case study method focusing on Cordoba (Spain). Although the subject is interesting, the paper needs extensive revisions as it looks there is no clear and fresh contribution to the studied area.

The choice of the subject needs more profound justification. The authors must explain why the relationship between visitor satisfaction and destination quality is essential in world heritage destinations listed in the UNESCO heritage list. Going more widely, what issues have been studied and explain concerning destination loyalty and destination image? There is such a vast area of research in tourism academia, every element of what the authors focus on was studied in details, analyzing the mediating roles of many other variables. The authors should clearly define the gap discussing at least the research problem focusing on the literature on world heritage destinations. I'm afraid I have to disagree with the authors stating that "This study is one of the first to analyze the relationship between tourism and expectations and contributes to the decisions which should be taken for the successful management of tourism in the city."

The authors present an unreflecting approach towards visitors satisfaction and their loyalty, focusing only on increasing the number of tourists as the effect of managerial decisions and actions on this matter. I would state that as long as the world heritage is concerned, the academic world has called for more sustainable strategies than boosterism. I hope the authors could join this discussion with their considerations and conclusions. As such, the literature part should be assessed as weak. Mentioned that, I cannot see the valuable contribution in the authors' statement that "The results of this study indicate that visitor satisfaction has a positive influence on loyalty to the destination, which encourages tourists to return to the destination in the future and to recommend it to others once they return to home".
The authors did not provide the research contribution they declare on page 2, line 57, when stating on reviewing of the variables that affect the loyalty of the heritage tourist. They do not refer to this in the discussion or conclusion part.

The authors should justify more the methodology. In the details, why do they adopt the American Customer Satisfaction Index (ACSI)? I am not against it, but the researcher should justify their choice. Why staying at least one week was used as the criterium of sample choice?

The papers present a weak discussion on the relationship between perceived value, visitors satisfaction, and visitors loyalty in the context of world heritage destinations in the light of the vast area of studies on the subject. The readers do not know what can they learn from Cordoba case study. I see the potential in discussing why H5 was not supported in the study. The authors should work more on the scientific soundness and the contribution of their paper.

The paper also needs proofreading; a lot of sentences are hard to understand.

Author Response

MANUSCRIP ID: ijerph-1001603: Loyalty in heritage tourism: the case of Córdoba and its four world heritage sites.

Dear editor and reviewers, thank you very much for allowing us revising and resubmitting tour article to the International Journal of Environmental Research and Public Health. We have found your comments to be highly helpful in improving the article in terms of the introduction, the theoretical background, the research methodology, and the theoretical contributions. After carefully reading your comments, we have introduced some changes in the manuscript to address your concerns. Following your recommendation, as shown in the new version of the manuscript. We present our detailed comments below.

RESPONSE TO REVIEWER 3

1) The article refers to a robust area of research on destination image destination, and especially the relationship between the quality assessment and satisfaction of visitors and destination loyalty in the context of world heritage sites. The authors used the case study method focusing on Cordoba (Spain). Although the subject is interesting, the paper needs extensive revisions as it looks there is no clear and fresh contribution to the studied area.

Thank you very much for your constructive suggestion, it is corrected. The authors have supplemented the penultimate paragraph in the "introduction" section in order to provide clarity to the subject matter of the article:

“…This study contributes to the literature and discusses the need for responsible tourism to contribute to the maintenance, development of heritage and the economy, especially the world heritage sites which are at risk from carbon footprints.”

2) The choice of the subject needs more profound justification. The authors must explain why the relationship between visitor satisfaction and destination quality is essential in world heritage destinations listed in the UNESCO heritage list. Going more widely, what issues have been studied and explain concerning destination loyalty and destination image? There is such a vast area of research in tourism academia, every element of what the authors focus on was studied in details, analyzing the mediating roles of many other variables. The authors should clearly define the gap discussing at least the research problem focusing on the literature on world heritage destinations. I'm afraid I have to disagree with the authors stating that "This study is one of the first to analyze the relationship between tourism and expectations and contributes to the decisions which should be taken for the successful management of tourism in the city."

We highly appreciate the feedback; En el penultimo párrafo del apartado introducción, hemos aclarado que el ambito de studio comprede a la única ciudad del mundo con cuatro inscripciones en el patrimonio mundial.

“…with four sites in the UNESCO list of world heritage sites.”

3) The authors present an unreflecting approach towards visitors satisfaction and their loyalty, focusing only on increasing the number of tourists as the effect of managerial decisions and actions on this matter. I would state that as long as the world heritage is concerned, the academic world has called for more sustainable strategies than boosterism. I hope the authors could join this discussion with their considerations and conclusions. As such, the literature part should be assessed as weak. Mentioned that, I cannot see the valuable contribution in the authors' statement that "The results of this study indicate that visitor satisfaction has a positive influence on loyalty to the destination, which encourages tourists to return to the destination in the future and to recommend it to others once they return to home".

Los autores presentan un enfoque sin refinar hacia la satisfacción de los visitantes y su lealtad, centrándose únicamente en el aumento del número de turistas como el efecto de las decisiones y acciones gerenciales en este asunto. Yo diría que mientras se trate del patrimonio mundial, el mundo académico ha pedido estrategias más sostenibles que el refuerzo. Espero que los autores puedan sumarse a este debate con sus consideraciones y conclusiones. Como tal, la parte de la literatura debe ser evaluada como débil. Mencionó que no veo la valiosa contribución en la declaración de los autores de que "los resultados de este estudio indican que la satisfacción del visitante tiene una influencia positiva en la lealtad al destino, lo que anima a los turistas a regresar al destino en el futuro y a recomendarlo a los demás una vez que regresen a casa".

Thank you very much for your constructive suggestion; The authors added the second paragraph in theorical background where we joined the debate on sustainability in sites with world heritage inscriptions.

2017 was designated as the International Year of Tourism for Sustainable Development. The purpose of UNWTO is to raise awareness of the role of tourism in the challenges posed by the effective use of resources and their effect on climate change [19-21]. Cultural heritage tourism should be an international reference for tourism at these destinations, which are especially sensitive to carbon footprints. The demand for sustainable tourism services by visitors who are concerned about SDGs increases day by day, as this type of tourist wants to know how to reduce their carbon footprint. By definition, the carbon footprint of tourism should include carbon emitted directly during tourist activities (combustion of gasoline in vehicles), as well as carbon incorporated into products purchased by tourists (food, accommodation, transport, fuel and shopping). UNWTO has defined sustainable tourism as Tourism that takes full account of its current and future economic, social and environmental impacts, meeting the needs of visitors, industry, the environment and host communities [22, 23]. The global tourism industry has also started implementing Tourism for SDGs, with the aim of contributing to Sustainable Development Goals (SDGs). This type of tourism is an alternative that improves the quality of life of the host community, provides a high-quality experience for visitors and maintains the quality of the environment on which both the host community and the visitor depend [24, 25].

4) The authors did not provide the research contribution they declare on page 2, line 57, when stating on reviewing of the variables that affect the loyalty of the heritage tourist. They do not refer to this in the discussion or conclusion part.

We highly appreciate your constructive comments, we've modified in the “discussion and conclusion” the pararaphe of the hypothetic 6.

“Finally, Hypothesis 6 was confirmed, which shows the direct influence and positive among the satisfaction of the heritage tourist with affect the loyalty of the heritage tourist. A high level of satisfaction is expected to have a positive effect on the loyalty of the heritage tourist to a heritage site. Based on previous research [80, 86, 87] the way in which the factors of satisfaction, quality and expectations influence the visitors’ intention to return to or recommend the destination were investigated. This study confirmed the relationship of high visitor satisfaction with the heritage of Cordoba and the willingness of tourists to return and recommend the city as a destination. This means that the loyalty of heritage tourists to a heritage destination is based on factors such as, safety at the destination and its surroundings, the accommodation, gastronomy, transport and the management of information provided to visitors, along with the comfort felt and the upkeep of the destination [88]. The visitor should feel that all this makes the tourist experience worth the price paid. In addition to enjoying visits to heritage sites, tourists also appreciate special rates, offers and discounts during their visit [89]”.

5). The authors should justify more the methodology. In the details, why do they adopt the American Customer Satisfaction Index (ACSI)? I am not against it, but the researcher should justify their choice. Why staying at least one week was used as the criterium of sample choice?

We highly appreciate your constructive comments, we have added paragraph seven in the "introduction" section:

“2017 was designated as the International Year of Tourism for Sustainable Development. The purpose of UNWTO is to raise awareness of the role of tourism in the challenges posed by the effective use of resources and their effect on climate change [19-21]. Cultural heritage tourism should be an international reference for tourism at these destinations, which are especially sensitive to carbon footprints. The demand for sustainable tourism services by visitors who are concerned about SDGs increases day by day, as this type of tourist wants to know how to reduce their carbon footprint. By definition, the carbon footprint of tourism should include carbon emitted directly during tourist activities (combustion of gasoline in vehicles), as well as carbon incorporated into products purchased by tourists (food, accommodation, transport, fuel and shopping). UNWTO has defined sustainable tourism as Tourism that takes full account of its current and future economic, social and environmental impacts, meeting the needs of visitors, industry, the environment and host communities [22, 23]. The global tourism industry has also started implementing Tourism for SDGs, with the aim of contributing to Sustainable Development Goals (SDGs). This type of tourism is an alternative that improves the quality of life of the host community, provides a high-quality experience for visitors and maintains the quality of the environment on which both the host community and the visitor depend [24, 25].”

6) The papers present a weak discussion on the relationship between perceived value, visitors satisfaction, and visitors loyalty in the context of world heritage destinations in the light of the vast area of studies on the subject. The readers do not know what can they learn from Cordoba case study. I see the potential in discussing why H5 was not supported in the study. The authors should work more on the scientific soundness and the contribution of their paper.

Thank you very much for the useful, which has helped us realizing the need to better explain unsupported hypotheses. We have more clearly detailed in the “discussion” section, (see five paragraph):

“The results of this study show that the perceived quality and the expected quality have a direct influence on satisfaction. The relationship of perceived value and satisfaction is not supported. One of the reasons for this is because satisfaction is influenced by the expectations created before the trip, the expectations that the visitor has after reading about the destination, planning itineraries, hotels, gastronomy, etc., and once the visitor is at the destination and has tried the gastronomy, decides on the level of satisfaction with the visit and recommends it if the level of satisfaction is high [82, 83]. The main reason why hypothesis 5 is not supported is due to the idiosyncrasies of this type of tourism. The results obtained coincide with studies that analyzed this type of tourism and the relationships between some of the factors analyzed in this study. It was found that the motivation of the tourist can influence their satisfaction and loyalty. When the motivation for the visit is due to cultural heritage, both the expectations and the expected quality for the visit take second place [84, 85] as the most important thing for the tourist motivated by cultural heritage is the visits to the sites, which alters the results for our hypotheses.”

Round 2

Reviewer 3 Report

I appreciate the effort the authors put in strengthening the paper, especially concerning the method. However, I would not state that the revised version is ready to publish. The theoretical part has not been strengthened enough. Moreover, it seems to be even weaker as the authors, instead of presenting the dynamics of the literature stream on consumer loyalty in heritage cities, focus on UWTO Sustainable Goals and associate the sustainability of tourism in heritage sites/cities with the carbon footprint. Well, it does not make sense as the carbon footprint impacts the travel sector as a part of global tourism. In heritage cities, other issues are important referring to sustainable goals, i.e. touristification, street pollution, overtourism, mass tourism, overcrowding, loss of the quality of heritage product, displacement of residents and traditional premises, gentrification etc. These elements concern other SD Goals and the visitor satisfaction and perceived value of the place than the carbon footprint. Moreover, the authors address them while discussing Hypothesis 5.
That is why I urge the actors to verify their statement due to improving the scientific soundness of the study.

Author Response

MANUSCRIP ID: ijerph-1001603: Loyalty in heritage tourism: the case of Córdoba and its four world heritage sites.

Dear editor and reviewers, thank you very much for allowing us revising and resubmitting tour article to the International Journal of Environmental Research and Public Health. We have found your comments to be highly helpful in improving the article in terms of the introduction, the theoretical background, the research methodology, and the theoretical contributions. After carefully reading your comments, we have introduced some changes in the manuscript to address your concerns. Following your recommendation, as shown in the new version of the manuscript. We present our detailed comments below.

RESPONSE TO REVIEWER 3

1) Requires revisions:

I appreciate the effort the authors put in strengthening the paper, especially concerning the method. However, I would not state that the revised version is ready to publish. The theoretical part has not been strengthened enough. Moreover, it seems to be even weaker as the authors, instead of presenting the dynamics of the literature stream on consumer loyalty in heritage cities, focus on UWTO Sustainable Goals and associate the sustainability of tourism in heritage sites/cities with the carbon footprint. Well, it does not make sense as the carbon footprint impacts the travel sector as a part of global tourism. In heritage cities, other issues are important referring to sustainable goals, i.e. touristification, street pollution, overtourism, mass tourism, Overcrowding, loss of the quality of heritage product, displacement of residents and traditional premises, gentrification etc. These elements concern other SD Goals and the visitor satisfaction and perceived value of the place than the carbon footprint. Moreover, the authors address them while discussing Hypothesis 5.

That is why I urge the actors to verify their statement due to improving the scientific soundness of the study.

We highly appreciate the feedback; it has led us to make substantial modifications to improve the Theorical background of the article. In particular, we've added issues like touristification, overcrowding, gentrification, as shown in the Theorical background and Discussion and Conclusion in the new version of the manuscript:

“2.1. Turistification, gentrification and tourismphobia

In recent years, the cities that are declared world heritage sites are undergoing a process of turistification. This process often has a negative impact with the depopulation of popular parts of the city which are in touristic areas [33, 34][35-40]. The depopulation of these areas brings with it the so-called gentrification process. After a process of urban transformation and the installation of new public security and cleaning services, these areas become fashionable and are demanded by tourists. Gentrification causes the displacement of the traditional population to other cheaper areas in the city [41-45]. This causes conflicts in the price of real estate, the disappearance of traditional trade and small artisans which are replaced by large franchises and companies prepared for mass tourism, new ambiguously regulated economic activities (Airbnb). These all lead to the displacement of traditional residents in touristic areas. Recent studies into gentrification and turistification analyze the interactions between the growth in tourism and the urban transformation of the tourist destination. In Spain, academic articles have been published about the sharing economy and the tourism industry in Madrid and Barcelona [33, 46, 47], but there are almost no contributions about other territories or smaller touristic cities that can be deeply affected, such as the city of Cordoba with its four World Heritage sites. Tourism strategies, that revitalize historic centers are created as a way to rescue abandoned or depressed places. However, if these policies are poorly applied without taking into account the load capacity of these places, the city can become overloaded and world heritage cities can be turned into "theme parks" [48, 49] which leads to problems of tourism management.

Scientific literature shows the negative effects of the above problem in big cities such as London, Paris, Prague, Madrid or Barcelona. The city of Barcelona in Spain attracts 76 million tourists per year, and there is now a new word in Spain, 'tourism-phobia,'. A law to curb tourism was passed in Venice, which has a total population of 55,000 people and an average of 60,000 visitors per day, with numbers increasing to 170,000 during Carnival. As a result, the residents of Venice are protesting about tourists with signs reading 'Tourists go away' or 'You are destroying this area'.[50-52]. On the other hand, in a large number of cases (medium-sized cities and developing countries), the inhabitants of these communities depend on tourism, which gives them economic gains, employment and an income, with improved infrastructures, services, and standard of living [41-43, 53, 54] These communities may prefer to support the social costs of heritage tourism.  [33-40, 44, 45].

Air pollution is another factor that can influence the decision to visit, the length of stay and even the rejection of destinations in World Heritage cities [55-58]. The city of Cordoba offers the visitor a wonderful opportunity to enjoy the city in this respect because it is a small city with little pollution, which is another necessary factor for a recommendable tourist destination of reference.

The most important World Heritage cities all have problems of massification [59-65] [61, 66, 67], O´Reilly (1986) [68] was the first to define the term "tourist capacity" as the maximum number of tourists that a destination can contain. This may reflect a new component in the concept of tourism, which promotes rational growth to avoid excesses and a negative impact on the environment [69-73].

Knowing about the consequences of tourist massification, gentrification and tourism-phobia can help the public administration and policy makers to provide strategies and take actions which can adapt to rapid changes in cities, avoid tourist bubbles [74], tourist massification and the depopulation of touristic parts of the city, avoiding the reduction in quality of heritage sites, enforcing responsible policies that balance the standard of living of residents with an adequate use of cultural heritage that contributes to the economic and social development of the city. This article is relevant in this context since it helps managers to take measures to properly manage tourism, taking into account the negative results that poor tourism management can have.”

In the Discussion and Conclusion relation to Hypothesis 5:

“Cultural and heritage motivation is so influential in this type of tourism that the research hypothesis of this study is not fulfilled as perceived value does not directly influence satisfaction. However, according to other authors, perceived value does influence other variables such as loyalty [121, 122]. Therefore, even though the hypothesis is not fulfilled, the variable should still be taken into account by public managers. This means that cities, that attract visitors because of important heritage sites, should research satisfaction rather than perceived value to establish a long-term relationship with visitors and tourists.”